# Midazolam impacts acetyl—And butyrylcholinesterase genes: An epigenetic explanation for postoperative delirium?

**Katharina Rump**◯*, **Caroline Holtkamp, Lars Bergmann, Hartmuth Nowak, Matthias Unterberg, Jennifer Orlowski, Patrick Thon, Zainab Bazzi, Maha Bazzi, Michael Adamzik, Björn Koos, Tim Rahmel**

Department of Anesthesiology, Intensive Care Medicine and Pain Therapy, University Hospital Knappschaftskrankenhaus Bochum, Ruhr-University Bochum, Bochum, Germany

* Katharina.k.rump@rub.de

**Data Availability Statement:** All relevant data are within the paper and its Supporting Information files.

## Abstract

Midazolam is a widely used short-acting benzodiazepine. However, midazolam is also criticized for its deliriogenic potential. Since delirium is associated with a malfunction of the neurotransmitter acetylcholine, midazolam appears to interfere with its proper metabolism, which can be triggered by epigenetic modifications. Consequently, we tested the hypothesis that midazolam indeed changes the expression and activity of cholinergic genes by acetylcholinesterase assay and qPCR. Furthermore, we investigated the occurrence of changes in the epigenetic landscape by methylation specific PCR, ChiP-Assay and histone ELISA. In an *in-vitro* model containing SH-SY5Y neuroblastoma cells, U343 glioblastoma cells, and human peripheral blood mononuclear cells, we found that midazolam altered the activity of acetylcholinesterase /buturylcholinesterase (AChE / BChE). Interestingly, the increased expression of the buturylcholinesterase evoked by midazolam was accompanied by a reduced methylation of the BCHE gene and the di-methylation of histone 3 lysine 4 and came along with an increased expression of the lysine specific demethylase KDM1A. Last, inflammatory cytokines were not induced by midazolam. In conclusion, we found a promising mechanistic link between midazolam treatment and delirium, due to a significant disruption in cholinesterase homeostasis. In addition, midazolam seems to provoke profound changes in the epigenetic landscape. Therefore, our results can contribute to a better understanding of the hitherto poorly understood interactions and risk factors of midazolam on delirium.

## 1. Introduction

Midazolam is the most abundantly used benzodiazepine in anesthesia and emergency medicine [1]. Due to its amnestic and anxiolytic effects, midazolam is considered as a favorable choice for premedication [2, 3]. However, the use of benzodiazepines especially midazolam is associated with postoperative complications such as cognitive impairment and delirium [4].

**Competing interests:** The authors have declared that no competing interests exist.

Currently, it is discussed whether anesthetics cause an alteration of the epigenetic landscape of the cell, which might induce a long-lasting cognitive impairment [5]. One common postoperative complication in elderly critically ill patients is the postoperative delirium (POD) that is also associated with a worse outcome, longer stay on the intensive care unit and higher healthcare related costs [6]. In addition, delirium is also linked to an increased risk of long term cognitive impairments that recover with high inter-individual differences from days to months [7]. Especially the use of benzodiazepine is, in addition to blood transfusion, one of the only modifiable factors with strong evidence for an association with delirium after surgery [8]. Within the group of benzodiazepines midazolam shows highest incidence of POD [9].

Although there are some theories that could explain the positive correlation between midazolam administration and the high incidence of POD, such as the degree of sedation [8] and the function of midazolam as a GABAergic agent [10], the underlying molecular mechanisms and the pathogenesis of POD still remain elusive.

Currently, a pathogenesis is discussed involving a reduced concentration of the neurotransmitter acetylcholine [11], neuroinflammation [12, 13] or decreased antiinflammation [14]. The hydrolysis of acetylcholine is mainly mediated by acetylcholinesterase (AChE) and butyrylcholinesterase (BChE) that can be found in the brain, red blood cells, and central nervous system [15]. Especially an altered activity and concentration of BChE seems to impact pathogenesis of POD [16–18] and BChE activity also shows high prognostic capability for POD [19].

Recently, we could demonstrate that the GABAergic agent propofol changes the epigenome [20]. In context with POD and anesthesia, the expression of lysine-specific demethylase (KDM1A) seems of special interest as it is associated with cognitive function [21] and demethylates histone 3 lysine 4 [22]. Hence long-lasting effects on the central nervous system and cognitive abilities caused by the GABAergic midazolam could be caused by changing the epigenetic landscape of the cells [23–25]. Since it is currently unknown whether and how midazolam influences the activity of the *ACHE* or *BCHE* gene. However, we speculate that one possible mechanism is the alteration of the expression of cholinergic genes by changing the epigenetic profile of the cells.

Therefore, in this study we investigated whether the expression, activity and methylation profile of cholinesterases are changed by midazolam. Furthermore, we study whether midazolam changes the epigenetic landscape of the cell by altering KDM1A expression.

## 2. Materials and methods

### 2.1. Cell culture

Human neuroblastoma cells SH-SY5Y and the glioblastoma cell line U343 (origin: Cell Lines Service, CLS, Eppelheim Germany, SH-SY5Y item number: 300154 and U343 item number: 300365) were cultured in Dulbecco's modified Eagle medium (DMEM; Gibco, Darmstadt, Germany) at 37°C and 5% $CO_2$ with 10% fetal calf serum (FCS; Gibco, Darmstadt, Germany) and 1% penicillin/streptomycin (Penstrep; Gibco, Darmstadt, Germany). Cells were maintained every three to four days by adding 5 ml of Trypsin-EDTA 0.25% (Gibco, Darmstadt, Germany) after medium removal to dissolve adhesive cells. Furthermore, peripheral blood mononuclear cells (PBMCs) were examined, after the Ethics Committee's approval (Ethics Committee of the Ruhr-University Bochum, Bochum, Germany; ref: 17-5964-BR), registration at the German Clinical Trials Register (ref: DRKS00012961, https://www.drks.de/drks_web/navigate.do?navigationId=trial.HTML&TRIAL_ID=DRKS00012961) and written informed consent. 80 ml EDTA blood was taken from eight healthy donors (5 female and 3 male) and PBMCs were isolated, using density gradient centrifugation with Ficoll-Paque (GE Healthcare, Chalfont, UK).

## 2.2. Quantitative reverse transcription PCR

q-RT-PCR on SH-SY5Y cells, U343 and PBMCs was performed as described previously [26]. Briefly, cells were cultured in 6-well culture plates and incubated with 250 ng/ml, 1 µg/ml or 50 µg/ml midazolam (midazolam hydrochloride injection solution, B. Braun Melsungen) for 2, 4 and 24 h, 10 µg/ml and 50 µg/ml flumazenil or were left untreated (control). Flumazenil incubation was performed two hours after starting midazolam incubation. For incubation, the highest concentration of midazolam (SH-SY5Y 50 µg/ml; U343 250 ng/ml; BV-2 10 µg/ml) and flumazenil was used, which did not reduce cell viability in different cell lines in previous experiments. Cells were incubated at 37˚C and 5% $CO_2$. After RNA isolation and cDNA synthesis of 1 µg RNA using the QuantiTect Reverse Transcription kit (Qiagen, Hilden, Germany), we utilized 2.5 µl of cDNA together with specific primers (Table 1) and GoTaq qPCR master mix (Promega, Madison, WI, USA) for a standard qPCR reaction protocol.

## 2.3. Cholinesterase activity after incubation with midazolam

Cholinesterase activity in SH-SY5Y cells was measured after stimulation with midazolam.

For this purpose, 5 x 105 SH-SY5Y cells were seeded in 4 ml of growth medium containing 10% FBS. Cells were incubated for 24 h at 37˚C and incubated for 2, 4 and 24 h with 50 µg/ml midazolam or were left untreated.

The proteins were isolated as previously described [20] after washing the cells with PBS. After the lysates were collected from all experiments, protein quantification was performed using the Rotiquant universal kit (Roth, Karlsruhe, Germany). The lysates were used for detection of cholinesterase activity using an acetylcholinesterase assay kit (fluorometric red) (Abcam, Cambridge, UK) according to the manufacturer's instructions.

**Table 1. Primer pairs for PCR.**

| Primer name | Sequence (5' to 3') | Product size (bp) |
|---|---|---|
| BCHE_M1_SE | ATTTAGGTTAAAACGGTGAAATTTC | 172 |
| BCHE_M1_AS | AAACTAAAATACCGTAACGCGAT | |
| BCHE_U1_SE | TTAGGTTAAAATGGTGAAATTTTGG | 173 |
| BCHE_U1_AS | CTCAAACTAAAATACCATAACACAAT | |
| ACHE_M_SE1 | AAT TTT ATT AGT TTC GAG CGA GAT C | 189 |
| ACHE_M_AS1 | GAC CCA AAA ACC TAC AAC GAC | |
| ACHE_U_SE1 | TTT TAT TAG TTT TGA GTG AGA TTG A | 188 |
| ACHE_U_AS1 | CAA CCC AAA AAC CTA CAA CAA C | |
| ACTB_SE | CTGGAACGGTGAAGGTGACA | 140 |
| ACTB_AS | AAGGGACTTCCTGTAACAATGCA | |
| KDM1A_RT_SE | GCCCACTTTATGAAGCCAACG | 161 |
| KDM1A_RT_AS | GCCAAGGGACACAGGCTTAT | |
| ACHE_mRNA_SE | GCT TCA GCA AAG ACA ACG AG | 115 |
| ACHE_mRNA_AS | GTG TAA TGC AGG ACC ACA GC | |
| BCHE_mRNA_SE | ATCCTGCATTTCCCCGAAGT | 239 |
| BCHE_mRNA_AS | CCGTGCCACCAAAAACTGTC | |
| BCHE_Prom_SE | GCATGTGCACTGCAAGTTGA | 90 |
| | AACTCTCGCGAGCTTTGTCA | |
| BCHE_Prom_AS | CCCTGCAGGCAGTCATACAT | |
| | CTGCTGCTCCAGCCTGTAAA | |

## 2.4. Methylation and expression of BCHE gene after incubation with midazolam

The DNA methylation of *BCHE* gene was quantified using methylation-specific PCR after bisulphite conversion in SH-SY5Y cells, before and after incubation. For this purpose, 5 x 105 SH-SY5Y cells per 4 ml were seeded in 6-well culture plates and incubated for 24 h at 37˚ C and 5% CO2. The cells were incubated with 50 μg/ml or 250 ng/ml midazolam depending on cell type for 2, 4 and 24h. Subsequently, the DNA was isolated using the QIAamp DNA blood mini kit (Qiagen, Hilden, Germany), following the manufacturer's instructions. Bisulphite conversion was performed with the EZ DNA methylation-gold kit (Zymo Research, Irvine, CA, USA). All DNA samples were diluted to 10 ng /μl qPCR was performed to detect methylation, as previously described [27], with the GoTaq qPCR master mix (Promega, Madison, WI, USA) and specific primers (Table 1).

The percentage of methylation was analyzed as previously described [27, 28].

## 2.5. Analysis of histone modifications

Furthermore, histone modifications of histone 3 after incubation were analyzed. SH-SY5Y cells and U343 were seeded, as previously described, and incubated with 250 ng / ml of midazolam or left untreated (control) for 24 h exactly as previously described [20].

Histone concentration was quantified by the Rotiquant universal kit (Roth, Karlsruhe, Germany) and histone modification was quantified by ELISA using 50 ng protein for the PathScan® Di-Methyl-Histone H3 (Lys4) Sandwich ELISA kit (Cell Signaling Technology, Cambridge, UK).

## 2.6. Chromatin immunoprecipitation assay (ChIP assay)

A ChIP assay was used to analyze if the promoter of the cholinergic gene *BCHE* binds to histone H3 lysine K4., 1 x $10^6$ SH-SY5Y were used for the Pierce agarose Chip kit (Thermo Fisher Scientific, Waltham, MA, USA). The H3K4me$_2$ polyclonal antibody (EpiGentek, Farmingdale, NY, USA) was used as a specific antibody. As a positive control, an antibody against RNA polymerase II in combination with specific primers against GAPDH was used, while Rabbit IgG in combination with our primers against *BCHE* gene regions was used as a negative control. After DNA isolation, PCR (One Taq Master Mix, New England Biolabs, Frankfurt am Main, Germany) was carried out with, *BCHE*_prom primers (Table 1), and the PCR products were analyzed on agarose gel (Peqlab, Erlangen, Germany).

## 2.7. LegendPlex assay for the quantification of cytokines (TNFα and IL6)

To measure cytokine release from glial cells, BV2 cells (kind gift from Veselin Grozdanov Department of Neurology, Ulm University, Ulm, Germany) were used. Cell culture supernatant was utilized after midazolam and LPS treatment for quantification of TNFα, IL6 with the Legend Plex InflammationPanel (BioLegend, San Diego, CA), according to manufacturer's recommendations. Briefly, cells were treated with 1 μg/mL midazolam 100 ng/ml LPS or left untreated and incubated for 2, 4 and 24 h in complete growth medium. Cell supernatant was collected and stored at -80˚C until use for cytokine quantification. Measurement was performed using FACS Canto II (Becton Dickinson GmbH, Heidelberg, Germany) according to the manufacturer's instructions and analysis was performed using LEGENDplex v8.0 software.

## 2.8. Statistics

All experiments were performed in duplicate and repeated at least three times. Results are presented as mean ± standard deviation. If not otherwise stated, all datasets were analyzed using

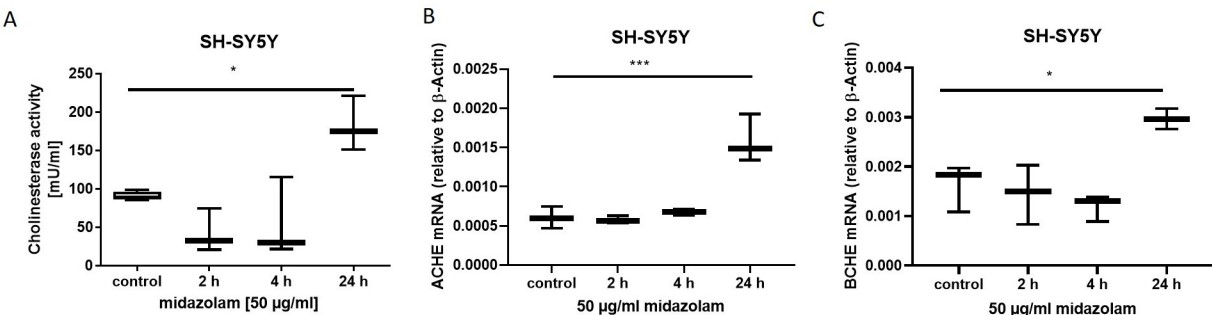

**Fig 1.** Activity and mRNA expression of AChE and BChE after incubation with midazolam A) the intracellular cholinesterase activity increased 24 h after midazolam exposure (n = 4; p = 0.01) and was measured by fluorometric assay B) *ACHE* mRNA quantified by qPCR expression was increased 24 hours after midazolam exposure in SH-SY5Y cells (n = 3; p<0.01) C) *BCHE* mRNA quantified by qPCR expression was increased 24 hours after midazolam exposure in SH-SY5Y cells (n = 3; p = 0.03). Data are presented as mean ± standard deviation. The reported p-value refers to the Dunnett's post-hoc test, comparing the underlying columns at the ends of each bar.

an unpaired t-Test or one-way ANOVA for multiple comparisons with a Dunnett's multiple comparisons test for specific comparisons. A p-value ≤ 0.05 was considered statistically significant. For multiple comparisons, specific comparisons were only analyzed if the one-way ANOVA showed a statistically significant difference between the groups. All statistical analyses were performed using GraphPad Prism 8 (San Diego, CA, USA).

## 3. Results

### 3.1. The activity and expression of AChE and BChE are altered after incubation with midazolam

Cholinesterase activity in SH-SY5Y cells was gradually reduced after incubation with midazolam, without reaching statistical significance (control: mean + SD 90.7 +5.5; 2 hours mean + SD 42.51 + 28.7; p = n.s.; Fig 1A) but the intracellular AChE and BChE activity (p = 0.01; Fig 1A) and *ACHE* (p< 0.01) and *BCHE* (p = 0.03) mRNA expression increased after 24 h by about 80% (Fig 1B and 1C).

### 3.2. Application of the midazolam antagonist flumazenil reverses midazolam induced effects on BCHE expression

In order to elucidate if midazolam antagonist flumazenil is capable to reduce midazolam induced effects on *ACHE* and *BCHE* expression, SH-SY5Y cells were incubated with flumazenil two hours after midazolam exposure. Here, we show a gradual increasing abolition of the midazolam effect (increased *BCHE* expression) under increasing doses of the antagonist flumazenil (Fig 2B). After the addition of 10μg/mL flumazenil the increased expression associated with midazolam of *BCHE* was reduced (p<0.05; Fig 2B). Interestingly, this effect was not observed on *ACHE* expression (Fig 2A).

### 3.3. Midazolam induces epigenetic changes in the BCHE gene of neuronal cells

Midazolam induced a decrease in *BCHE* intron 2 DNA methylation (p = 0.01; Fig 3A) and in the di-methylation of H3K4 (p = 0.02; Fig 3B), where *BCHE* promoter binds (Fig 3C). *ACHE* DNA methylation was not altered by incubation with 50 μg/ml midazolam (Fig 3D).

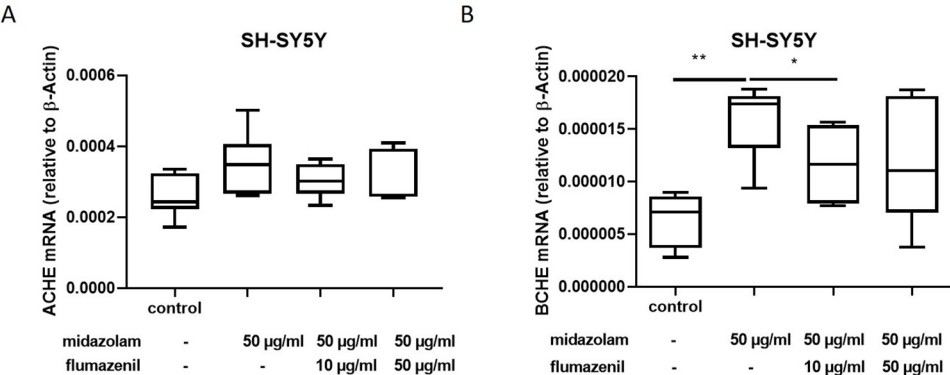

**Fig 2. SH-SY5Y cells were incubated with midazolam for 24 h and with flumazenil (starting 2 h after midazolam exposure) for 22 hours.** ACHE and BCHE mRNA expression relative to β-Actin mRNA expression were quantified by qPCR. Incubation with flumazenil A) did not alter ACHE expression (n = 6; p = n.s.) and reduced B) BCHE (n = 6; p = 0.046) expression. Data are presented as mean ± standard deviation. The reported p-value refers to the Dunnett's post-hoc test, comparing the underlying columns at the ends of each bar.

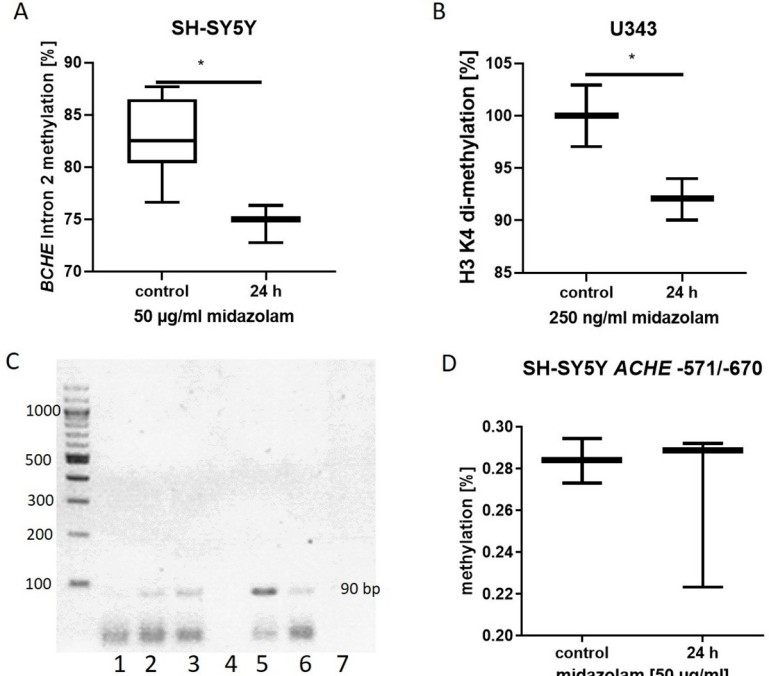

**Fig 3.** Methylation of *BCHE* in neuronal SH-SY5Y and U343 cells after midazolam incubation A) *BCHE* intron 2 methylation reduced after midazolam (50 μg/ml) exposure of SH-SY5Y cells (n = 3; p = 0.01) analyzed by methylation specific PCR. B) ELISA showed that histone H3 lysine 4 di-methylation (H3K4me2) decreased in U343 cells after incubation with 250 ng/ml midazolam (n = 3; p = 0.02) C) Chip-Assay confirmed binding of *BCHE* promoter region (90 bp) to H3K4me2; a 100 bp DNA Ladder was utilized; lanes 1, 7 show incubation with H3K27 antibody; lanes 2 and 6 show incubation with H3K4 antibody; lane 4 shows negative control without antibody and lane 5 shows positive control with RNA-polymerase II antibody (two experiments out of three are shown; n = 3). D) ACHE -571/-670 promoter methylation was not affected by midazolam (50 μg/ml) exposure of SH-SY5Y cells (n = 3; p = n.s.) analyzed by methylation specific PCR. Data are presented as mean ± standard deviation.

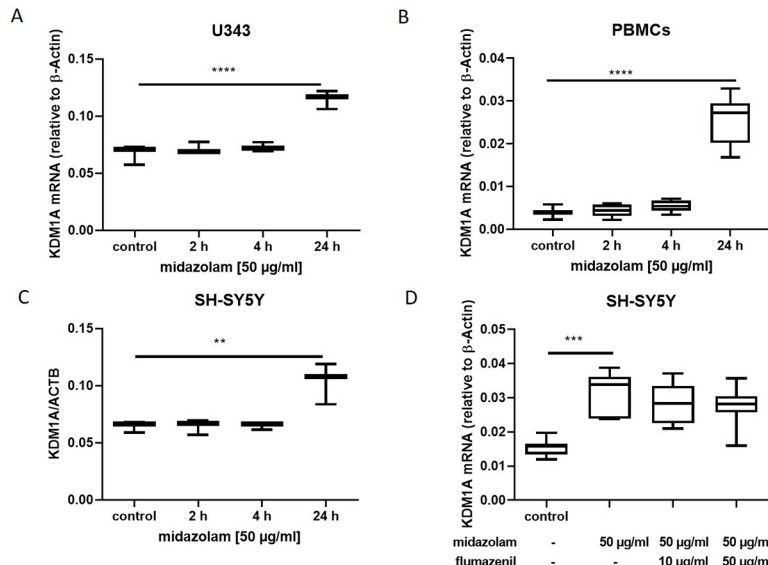

**Fig 4. KDM1A mRNA expression was quantified relative to β-Actin mRNA expression by qPCR.** Increased expression of lysine specific demethylase (*KDM1A*) in different cells after midazolam [50 μg/ml] exposure for 24 hours analyzed by qPCR. *KDM1A* expression increased in U343 (n = 3; A, in peripheral blood mononuclear cells (PBMCs) (n = 8; B) and in SH-SY5Y (n = 3; C). Flumazenil did not reduce KDM1A expression (n = 6, D). Data are presented as mean ± standard deviation. The reported p-value refers to the Dunnett's post-hoc test, comparing the underlying columns at the ends of each bar.

## 3.4. Midazolam increases the expression of lysine specific demethylase KDM1A

To explore the underlying mechanisms for the decrease in H3K4me2, we analyzed the expression of lysine specific demethylase *KDM1A* after exposure to midazolam. *KDM1A* mRNA expression was increased in, in U343 by about 50% (p <0.01; Fig 4A), in PBMCs by more than 100% (p<0.01; Fig 4B) and in SH-SY5Y by about 50% (p = 0.0038; Fig 4C). Incubation with flumazenil reduced midazolam induced effects in a visible dose dependent manner in SH-SY5Y cells, while incubation with midazolam alone led to increased expression of KDM1A (p < 0.001; Fig 4D) expression.

## 3.5. Midazolam does not induce the release of cytokines from BV-2 glial cells

Since postoperative delirium is strongly associated with neuroinflammation, we finally investigated whether midazolam itself evoked cytokine secretion in neural glial cells (BV-2, RRID: CVCL_0182). Midazolam did not induce any change in cytokine secretion in BV-2 cells (p = ns), compared to untreated cells. Cells incubated with lipopolysaccharide (LPS) as positive control had higher TNF-α cytokine levels (p = 0.01; Fig 5A) and higher IL-6 levels (p = 0.02; Fig 5B) compared to cells incubated with midazolam for 24 hours.

## 4. Discussion

Midazolam is a widely used benzodiazepine although its application is associated with the occurrence of POD [9]. A potential mechanism for the development of delirium is impaired cholinergic transmission based on the deficiency of acetylcholine in the brain [29]. However, as the causal relationship between midazolam and the cholinergic system is unknown, we systematically analyzed the expression and epigenetic regulation of cholinergic genes in neuronal

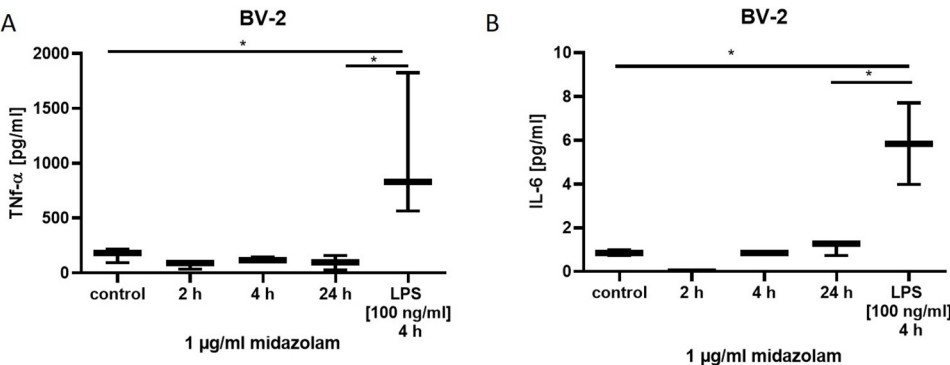

**Fig 5. Cytokine secretion in BV-2 glial cells after midazolam (1μg/ml) for 24 hours and LPS (100 ng/ml) for 4 hours (n = 3).** BV-2 cells were incubated with midazolam (1μg/ml) or lipopolysaccharide LPS (100 ng/ml) or left untreated. Cytokine expression was quantified using a bead-based immunoassay. Data are presented as mean ± standard deviation. The reported p-value refers to the Dunnett's post-hoc test, comparing the underlying columns at the ends of each bar.

cells after midazolam exposure. As a different postoperative activity of the proteins AChE and BChE in patients is already described [16, 30, 31] it seems of special interest, how their gene expression is regulated after midazolam exposure.

First, we could detect a visibly early decrease in cholinesterase activity and a slight decrease in the expression of *BCHE* mRNA, but a late increase in the activity and the expression of *ACHE* and *BCHE* mRNA. Our results regarding AChE and BChE activity and expression are in line with other studies analyzing AChE and BChE activities in peripheral blood from preoperative and postoperative patients [16, 19, 30]. AChE and BChE concentrations in blood and cerebrospinal fluid were altered in patients undergoing total hip/knee replacement, and BChE concentration showed the highest prognostic value for the development of POD [19]. Thus, increased gene expression, especially BChE, could represent an important mechanism, as it could be found in the brain of patients with Alzheimer's disease [32] and several studies explored the therapeutic implication of cholinesterase inhibitors in alleviating postoperative delirium [33].

Second, we tested the methylation of a *BCHE* gene region and a histon, with *BCHE* binding affinity. Methylation of the *BCHE* gene region (Intron 2) and the H3K4 di-methylation decreased after midazolam incubation. Thus, it seems appropriate to suggest that the region of the *BCHE* gene we investigated has activating effects on the transcription of this gene. However, it must be mentioned that the reduction in methylation was only about 10%. This seems to be questionable for a more than doubled amount of mRNA expression. In fact, other studies have already shown that a small change in DNA methylation of approximately 5% can have a great impact on gene expression [34]. Therefore, it seems possible that this small change in methylation state may cause this effect on mRNA expression.

Third: Since midazolam changed the di-methylation of H3K4, and we could detect binding of *BCHE* to this histone, it seems appropriate that midazolam might change the epigenome of the cell by influencing histon-modifying enzymes. H3K4me2 has been shown to mark actively transcribing genes [35]. In our analyzed neuronal cell line di-methylation was nearly 100 percent and midazolam could decrease the methylation slightly. A reduction of the di-methylation of H3K4 could therefore mean an overall increase in BChE expression. The demethylation of H3K4 is facilitated by KDM1A and is a well-established mechanism underlying transcriptional gene repression, but recently its role in gene activation could be shown [36]. The KDM1A demethylation of H3K4me2 in GR-targeted enhancers was shown to be important for GC-

mediated gene transcription, facilitating a molecular mechanism for the demethylation of H3K4me2 in gene activation [36]. Since changes in the methylation of histone 3 is facilitated by KDM1A, we analyzed the expression of *KDM1A* in our cell lines, and because POD is also associated with an altered cholinesterase activity in blood samples [16], we additionally investigated the expression of these enzymes in PBMCs. Strikingly, *KDM1A* showed a significant increased expression after midazolam treatment in all investigated cell lines (including PBMCs). Thus, our results provide first evidence that midazolam indeed rewrites the epigenetic landscape of the cell. Interestingly, the application of KDM1A inhibitors is associated with positive effects on memory. Recently it could be demonstrated that inhibition of KDM1A corrects memory deficit and behavior alterations in a mouse model of Alzheimer's Disease [21]. Another KDM1A inhibitor T-448 improved learning function in mice suffering from neuronal glutamate receptor hypofunction [37]. Thus, it seems tempting to speculate that KDM1A inhibitors might represent a therapeutic approach against POD. However, this crude thesis needs to be evaluated in upcoming studies.

Fourth: As increased expression of *BCHE* seems to be critical mechanisms after midazolam exposure. In this context, we analyzed if the midazolam antagonist flumazenil could inhibit midazolam induced effects. Indeed, we could show that flumazenil application reduced midazolam-induced expression in a dose-dependent manner. Regarding the effects of flumazenil application after midazolam anesthesia on brain function, we can only speculate. However, it is known that cognitive abnormalities can significantly be ameliorated after benzodiazepine use by slow subcutaneous infusion of flumazenil [38] and that flumazenil administration attenuates cognitive impairment [38]. Therefore, flumazenil use might be effective in reducing POD.

Lastly, since POD is related to neuroinflammation [39], we analyzed if there is a link between midazolam treatment for neuroinflammation. We could demonstrate that in our positive control, the incubation of glial cells with LPS TNF-alpha and IL-6 were significantly upregulated. However, midazolam treatment had no influence on the expression of these cytokines. IL-6 seems to be of particular interest as it seems to be a consistent predictor of delirium in surgical samples [40]. Therefore, we can conclude that midazolam does not strongly contribute to pro-inflammatory signaling, being discussed as additional factors in the development of POD [12–14].

We have to discuss the limitations of our study. Direct transfer to the bedside is inappropriate because we worked with cell lines as a model for the human brain. However, for instance we chose the neuronal cell line SH-SY5Y, because these represent an established cell line used to study brain disorders such as Alzheimer's disease or Parkinson [41, 42]. In addition, the extraction of neuronal cells from healthy volunteers or patients with POD is ethically not feasible [43]. Despite great efforts made to achieve the highest possible degree of standardization, variance in effect sizes or observed effects can occur within the individual experiments, which limits the statistical or mathematical accuracy of our experiments. However, this had no or only a negligible effect on the interpretation of our data. Therefore, considering the limitations of immortalized cell lines, we are confident that it is appropriate to perform our investigations in our selected cell lines. In addition, direct measurement of acetylcholine would be interesting but is not suitable as it is extremely unstable [44]. Thus, we mainly refer to the central effectors and regulators of acetylcholine concentration.

## 5. Conclusions

In summary, we found that midazolam upregulates intracellular BCHE expression. This upregulation in expression might be caused by demethylation of BCHE gene and H3K4 me2

demethylation and be facilitated by KDM1A. Thus, our results underpin the thesis, that over-expression of BCHE might aggravate postoperative delirium, due to an increased hydrolysis of acetyl-choline. Although POD is closely related to neuroinflammation, midazolam appears to be a separate trigger, independent of inflammation. Further studies should validate our promising results and mechanistic implications in the clinical context regarding feasibility and transferability.

## Supporting information

**S1 Raw images.**
(JPG)

## Author Contributions

**Conceptualization:** Katharina Rump, Hartmuth Nowak.

**Data curation:** Katharina Rump, Caroline Holtkamp, Tim Rahmel.

**Formal analysis:** Katharina Rump, Lars Bergmann, Jennifer Orlowski, Patrick Thon, Björn Koos.

**Investigation:** Katharina Rump, Caroline Holtkamp, Matthias Unterberg.

**Methodology:** Katharina Rump, Caroline Holtkamp, Hartmuth Nowak, Matthias Unterberg, Jennifer Orlowski, Patrick Thon, Zainab Bazzi, Maha Bazzi, Tim Rahmel.

**Project administration:** Michael Adamzik, Björn Koos, Tim Rahmel.

**Supervision:** Lars Bergmann, Hartmuth Nowak, Michael Adamzik, Tim Rahmel.

**Writing – original draft:** Katharina Rump.

**Writing – review & editing:** Caroline Holtkamp, Lars Bergmann, Hartmuth Nowak, Matthias Unterberg, Michael Adamzik, Björn Koos, Tim Rahmel.

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
