## [Decision Letter · Decision Letter 0]

2 Jun 2022

PONE-D-22-13550Midazolam impacts acetyl- and butyrylcholinesterase genes: An epigenetic explanation for postoperative delirium?PLOS ONE

Dear Dr. Rump,

Thank you for submitting your manuscript to PLOS ONE. After careful consideration, we feel that it has merit but does not fully meet PLOS ONE’s publication criteria as it currently stands. Therefore, we invite you to submit a revised version of the manuscript that addresses the points raised during the review process. In your revised manuscript please address the comments of both reviewers as fully as possible.

We look forward to receiving your revised manuscript.

Kind regards,

Israel Silman

Academic Editor

PLOS ONE

Journal Requirements:

2.

"NO"

4. PLOS ONE now requires that submissions reporting blots or gels include original, uncropped blot/gel image data as a supplement or in a public repository. This is in addition to complying with our image preparation guidelines described at https://journals.plos.org/plosone/s/figures#loc-blot-and-gel-reporting-requirements. These requirements apply both to the main figures and to cropped blot/gel images included in Supporting Information. If the manuscript is positively reviewed, we will ask the authors to provide any missing raw image data for blot/gel results when they submit their first revision. As part of your review, please ensure that figures reporting blot or gel images comply with the journal’s image preparation guidelines and that the original data are provided following the journal’s request.  If you have any questions or concerns about blot/gel figures or data for this submission, please email us at plosone@plos.org before issuing a decision letter.

Reviewers' comments:

Reviewer's Responses to Questions

**Comments to the Author**

1. Is the manuscript technically sound, and do the data support the conclusions?

Reviewer #1: Yes

Reviewer #2: Partly

2. Has the statistical analysis been performed appropriately and rigorously? 

Reviewer #1: Yes

Reviewer #2: Yes

3. Have the authors made all data underlying the findings in their manuscript fully available?

Reviewer #1: Yes

Reviewer #2: Yes

4. Is the manuscript presented in an intelligible fashion and written in standard English?

Reviewer #1: Yes

Reviewer #2: Yes

5. Review Comments to the Author

Reviewer #1: Summary: Midazolam is given to patients before surgery. Midazolam is responsible for the delirium experienced by some patients in the days after surgery. Based on previous reports that cholinesterase activity levels are altered in patients who experience delirium, the present work hypothesized that midazolam affects the cholinesterase genes. Stable cell lines and peripheral blood mononuclear cells were assayed for the effect of midazolam on cholinesterase activity and mRNA levels, methylation of cholinesterase genes, demethylation enzyme activity, binding of the BCHE promoter to Histone and neuroinflammation. It was concluded that BCHE gene overexpression may aggravate postoperative delirium.

1. Abstract. Please mention that your studies used human SH-SY5Y neuroblastoma cells, U343 glioblastoma cells, and human peripheral blood mononuclear cells.

2. Abstract. Please list the methods used in your study.

3. You find that midazolam increased the level of BChE activity. However, references 16 and 18 report decreased BChE activity in the blood of patients treated with midazolam, on day 1 after surgery. Perhaps the abstract refers to increased BCHE mRNA rather than BChE activity.

4. Page 4 of 20, line 80. Typing error: 2.5% should be 0.25% Trypsin-EDTA

5. Page 4 of 20. Did you wash the cells with phosphate buffer saline before adding Trypsin-EDTA? This question is relevant to your assay of AChE activity because fetal bovine serum has substantial AChE activity. If you did not wash the cells, bovine AChE activity may be included in the AChE activity reported for cell lysates.

6. Please state the source of midazolam. Which midazolam salt did you use? What solvent was used to dissolve midazolam?

7. Page 4 of 20, line 94. Please define the abbreviation BV-2.

8. The word “stimulated” is used throughout the text when describing treatment with midazolam. What observation explains your choice of the word “stimulated”. Perhaps the cells proliferated more rapidly in the presence of midazolam?

9. Figure 1A. The y axis is labeled AChE/BChE activity, indicating AChE activity was divided by BChE activity. However, the assay measured total AChE and BChE activity because both enzymes hydrolyze the same substrate. Please change the y axis to Cholinesterase activity.

10. Figures 1B and 1C. The label on the y axis gives no clue that the assay measures mRNA level. Please modify the label on the y axis to AChE mRNA and BChE mRNA

11. The abbreviation ACTB is on the y axis of Figures 1B, 1C, , 2A, 2B, 4A, 4B, 4C, and 4D. Please replace ACTB with a word that is easy to understand. Perhaps the ACTB label is intended to mean that the reported mRNA quantity is relative to the mRNA for beta actin. This should be explained in the figure legend.

12. Page 8 of 20, line 178. It is suggested to write “the midazolam antagonist flumazenil reverses midazolam induced effects”

13. Figure 2 is missing the letters A and B for panels A and B.

14. Figure 2A please change the label on the y axis to AChE mRNA

15. Figure 2B please change the label on the y axis to BChE mRNA

16. Page 9 of 20. Line 186. Typing error Figure 1A should be Figure 2A.

17. Page 9 of 20, lines 194 to 196. Typing error. Figures 2A, 2B, 2C should be Figures 3A, 3B, 3C.

18. Figure 3 legend. The words lines and line should be lanes and lane.

19. Figure 3C. Please add numbers for DNA sizes. Of special interest are the sizes of the bands in lanes 1 to 7.

20. Page 10 of 20, lines 226-227. Typing error. Figures 4A and 4B should be 5A and 5B.

21. Conclusion. Line 323. BChE activity is extremely low to almost undetectable in stable cell lines. Your assay measured total cholinesterase activity, which in effect is AChE activity. Please delete the word “activity” in line 323 because BChE activity was not measured.

Reviewer #2: This interesting study pursued the epigenetic origin of delirium by focusing on methylation changes in the butyrylcholinesterase(BChE) promoter and seeking the corresponding methylase changes in blood cells from patients treated with Midazolam and in cultured human-originated neuroblastoma cells.The concept is a strength point of this study, but there are weakness points as well, as listed below.

1. Why limit the search for methylation but avoid pursuit of microRNAs targeting the cholinesterase mRNA transcripts?

2. What is the sex origin of the studied cells and what about the other sex? Pursuing a regulation process in half of humanity seems odd.

3. Rather than presenting bar graphs, please shift the graphs into box plots displaying the variability.

6. PLOS authors have the option to publish the peer review history of their article (what does this mean?). If published, this will include your full peer review and any attached files.

Reviewer #1: No

Reviewer #2: No

---

## [Author Response · Author response to Decision Letter 0]

21 Jun 2022

Reviewer #1: Summary: Midazolam is given to patients before surgery. Midazolam is responsible for the delirium experienced by some patients in the days after surgery. Based on previous reports that cholinesterase activity levels are altered in patients who experience delirium, the present work hypothesized that midazolam affects the cholinesterase genes. Stable cell lines and peripheral blood mononuclear cells were assayed for the effect of midazolam on cholinesterase activity and mRNA levels, methylation of cholinesterase genes, demethylation enzyme activity, binding of the BCHE promoter to Histone and neuroinflammation. It was concluded that BCHE gene overexpression may aggravate postoperative delirium.

In response: We are pleased that you consider our manuscript to address an important topic, and we are happy to discuss the concerns you raised. Please find below a point-by-point reply addressing your remarks.

1. Abstract. Please mention that your studies used human SH-SY5Y neuroblastoma cells, U343 glioblastoma cells, and human peripheral blood mononuclear cells.

In response: Thank you very much for this helpful comment. We now included into the abstract in lines 20 and 21: 

In an in-vitro model containing SH-SY5Y neuroblastoma cells, U343 glioblastoma cells, and human peripheral blood mononuclear cells, we found that midazolam altered the activity of acetylcholinesterase /buturylcholinesterase (AChE / BChE).

2. Abstract. Please list the methods used in your study.

In response: Thank you very much for this helpful comment. We now included into the abstract in lines 18 till 20: 

Consequently, we tested the hypothesis that midazolam indeed changes the expression and activity of cholinergic genes by acetylcholinesterase assay and qPCR. Furthermore, we investigated the occurrence of changes in the epigenetic landscape by methylation specific PCR, ChiP-Assay and histone ELISA.

3. You find that midazolam increased the level of BChE activity. However, references 16 and 18 report decreased BChE activity in the blood of patients treated with midazolam, on day 1 after surgery. Perhaps the abstract refers to increased BCHE mRNA rather than BChE activity.

In response: Thank you again for your valuable comment. Indeed, it is described in literature that BChE activity is decreased 48 hours after surgery, while AChE activity is not affected. In our study we could not differentiate between acetylcholesterase (AchE) or butyrylcholinesterase (BChE) activity as both enzymes can hydrolyze acetylcholine. However, we saw that BCHE mRNA expression slightly decreased after midazolam exposure, which was followed by an increase after 24 hours. Hence, this could depict a feedback mechanism for the initial decrease. 

4. Page 4 of 20, line 80. Typing error: 2.5% should be 0.25% Trypsin-EDTA

In response: We corrected this typing error.

5. Page 4 of 20. Did you wash the cells with phosphate buffer saline before adding Trypsin-EDTA? This question is relevant to your assay of AChE activity because fetal bovine serum has substantial AChE activity. If you did not wash the cells, bovine AChE activity may be included in the AChE activity reported for cell lysates.

In response: Thank you very much for this comment. Yes, cell lysate preparation for AChE/BChE activity assay contained a wash step with PBS. We now included in line 111: 

The proteins were isolated as previously described (20) after washing the cells with PBS.

6. Please state the source of midazolam. Which midazolam salt did you use? What solvent was used to dissolve midazolam?

In response: We utilized midazolam hydrochloride injection solution from B. Braun, Melsungen. It is dissolved in 10 % HCl and sterile water. We now included in line 94: 

Briefly, cells were cultured in 6-well culture plates and incubated with 250 ng/ml, 1 µg/ml or 50 µg/ml midazolam (midazolam hydrochloride injection solution, B. Braun Melsungen) for 2, 4 and 24 h, 10 µg/ml and 50 µg/ml flumazenil or were left unstimulated (control).

7. Page 4 of 20, line 94. Please define the abbreviation BV-2.

In response: I am sorry for not providing you an abbreviation for this cell line. However, the accession number was now included in the test on line 229: neural glial cells (BV-2, RRID:CVCL_0182 )

8. The word “stimulated” is used throughout the text when describing treatment with midazolam. What observation explains your choice of the word “stimulated”. Perhaps the cells proliferated more rapidly in the presence of midazolam?

In response: Thank you for this comment. I agree that “stimulate” is not neutral wording but implies a change in the cells biochemical processes. Indeed, we incubated the cells to examine putative stimulation. I changed my language and replaced the word “stimulated” by “incubated” or “treated” throughout the whole manuscript. 

9. Figure 1A. The y axis is labeled AChE/BChE activity, indicating AChE activity was divided by BChE activity. However, the assay measured total AChE and BChE activity because both enzymes hydrolyze the same substrate. Please change the y axis to Cholinesterase activity.

Figure 1 has been changed. 

10. Figures 1B and 1C. The label on the y axis gives no clue that the assay measures mRNA level. Please modify the label on the y axis to AChE mRNA and BChE mRNA

Figure 1 has been changed.

11. The abbreviation ACTB is on the y axis of Figures 1B, 1C, , 2A, 2B, 4A, 4B, 4C, and 4D. Please replace ACTB with a word that is easy to understand. Perhaps the ACTB label is intended to mean that the reported mRNA quantity is relative to the mRNA for beta actin. This should be explained in the figure legend.

The figures were changed and the legends of Figure 2 and 4 were modified. 

12. Page 8 of 20, line 178. It is suggested to write “the midazolam antagonist flumazenil reverses midazolam induced effects”

This was changed.

13. Figure 2 is missing the letters A and B for panels A and B.

We changed this. 

14. Figure 2A please change the label on the y axis to AChE mRNA

15. Figure 2B please change the label on the y axis to BChE mRNA

We changed this. 

16. Page 9 of 20. Line 186. Typing error Figure 1A should be Figure 2A.

We changed this.

17. Page 9 of 20, lines 194 to 196. Typing error. Figures 2A, 2B, 2C should be Figures 3A, 3B, 3C.

We changed this.

18. Figure 3 legend. The words lines and line should be lanes and lane.

We changed this.

19. Figure 3C. Please add numbers for DNA sizes. Of special interest are the sizes of the bands in lanes 1 to 7.

We added the numbers for DNA size. 

20. Page 10 of 20, lines 226-227. Typing error. Figures 4A and 4B should be 5A and 5B.

We changed this.

21. Conclusion. Line 323. BChE activity is extremely low to almost undetectable in stable cell lines. Your assay measured total cholinesterase activity, which in effect is AChE activity. Please delete the word “activity” in line 323 because BChE activity was not measured.

We changed this.

Again, thank you very much for carefully reading the manuscript. All points you raised were considered and the errors were corrected. 

Reviewer #2: This interesting study pursued the epigenetic origin of delirium by focusing on methylation changes in the butyrylcholinesterase(BChE) promoter and seeking the corresponding methylase changes in blood cells from patients treated with Midazolam and in cultured human-originated neuroblastoma cells.The concept is a strength point of this study, but there are weakness points as well, as listed below.

In response: We are pleased that you consider our manuscript to address an important topic, and we are happy to discuss the concerns you raised. Please find below a point-by-point reply addressing your remarks.

1. Why limit the search for methylation but avoid pursuit of microRNAs targeting the cholinesterase mRNA transcripts?

In response: Thank you very much for this outstanding comment. We agree that microRNA has a great potential to regulate cholinesterase mRNA expression. However, the search term “cholinesterase microRNA” yields 70 results in pub med. Hence, we conclude that it depicts a relatively good studied mechanism, while to our knowledge methylation of cholinergic genes was not studied at all. We therefore focused on DNA methylation analysis, but will include studying microRNA in our future studies regarding midazolam and cholinergic activity. 

2. What is the sex origin of the studied cells and what about the other sex? Pursuing a regulation process in half of humanity seems odd.

In response: Thank you very much for this interesting comment. We mainly studied SH-SY5Y cells, which are from a female origin. In addition, U-343MG cells were studied, which are from a male origin. To completely circumvent gender specific effects, we utilized PBMCs from healthy controls (5 female and 3 male probands), where no gender dependent effect could be seen. We now included gender distribution into the methods section in line 88 on page 4. 

3. Rather than presenting bar graphs, please shift the graphs into box plots displaying the variability.

In response: Thank you very much for this outstanding comment. We now changed all graphs. 

In addition, we have now added new data into figure 3 d, where in the previous version “data not shown” was stated.

---

## [Editor Report · Decision Letter 1]

24 Jun 2022

Midazolam impacts acetyl- and butyrylcholinesterase genes: An epigenetic explanation for postoperative delirium?

PONE-D-22-13550R1

Dear Dr. Rump,

We’re pleased to inform you that your manuscript has been judged scientifically suitable for publication and will be formally accepted for publication once it meets all outstanding technical requirements.

Kind regards,

Israel Silman

Academic Editor

PLOS ONE
---

## [Editor Report · Acceptance letter]

29 Jun 2022

PONE-D-22-13550R1 

Midazolam impacts acetyl- and butyrylcholinesterase genes: An epigenetic explanation for postoperative delirium? 

Dear Dr. Rump:

I'm pleased to inform you that your manuscript has been deemed suitable for publication in PLOS ONE. Congratulations! Your manuscript is now with our production department. 

Kind regards, 

on behalf of

Prof. Israel Silman 

Academic Editor

PLOS ONE